# BEVDISTILL: CROSS-MODAL BEV DISTILLATION FOR MULTI-VIEW 3D OBJECT DETECTION

**Zehui Chen**[1], **Zhenyu Li**[2], **Shiquan Zhang**[3], **Liangji Fang**[3], **Qinhong Jiang**[3], **Feng Zhao**[1*]

[1] University of Science and Technology of China
[2] Harbin Institute of Technology
[3] SenseTime Research

`lovesnow@mail.ustc.edu.cn, fzhao956@ustc.edu.cn`
`zhenyuli17@hit.edu.cn`
`{zhangshiquan,fangliangji,jiangqinhong}@senseauto.com`

## ABSTRACT

3D object detection from multiple image views is a fundamental and challenging task for visual scene understanding. Owing to its low cost and high efficiency, multi-view 3D object detection has demonstrated promising application prospects. However, accurately detecting objects through perspective views is extremely difficult due to the lack of depth information. Current approaches tend to adopt heavy backbones for image encoders, making them inapplicable for real-world deployment. Different from the images, LiDAR points are superior in providing spatial cues, resulting in highly precise localization. In this paper, we explore the incorporation of LiDAR-based detectors for multi-view 3D object detection. Instead of directly training a depth prediction network, we unify the image and LiDAR features in the Bird-Eye-View (BEV) space and adaptively transfer knowledge across non-homogenous representations in a teacher-student paradigm. To this end, we propose **BEVDistill**, a cross-modal BEV knowledge distillation (KD) framework for multi-view 3D object detection. Extensive experiments demonstrate that the proposed method outperforms current KD approaches on a highly-competitive baseline, BEVFormer, without introducing any extra cost in the inference phase. Notably, our best model achieves 59.4 NDS on the nuScenes test leaderboard, achieving new state-of-the-arts in comparison with various image-based detectors. Code will be available at https://github.com/zehuichen123/BEVDistill.

## 1 INTRODUCTION

3D object detection, aiming at localizing objects in the 3D space, is a crucial ingredient for 3D scene understanding. It has been widely adopted in various applications, such as autonomous driving (Chen et al., 2022a; Shi et al., 2020; Wang et al., 2021b), robotic navigation (Antonello et al., 2017), and virtual reality (Schuemie et al., 2001). Recently, multi-view 3D object detection has drawn great attention thanks to its low cost and high efficiency. As images offer a discriminative appearance and rich texture with dense pixels, detectors can easily discover and categorize the objects, even in a far distance. Despite the promising deployment advantage, accurately localizing instances from camera view only is extremely difficult, mainly due to the ill-posed nature of monocular imagery. Therefore, recent approaches adopt heavy backbones (*e.g.,* ResNet-101-DCN (He et al., 2016), VoVNetV2 (Lee & Park, 2020)) for image feature extraction, making it inapplicable for real-world applications.

LiDAR points, which capture precise 3D spatial information, provide natural guidance for camera-based object detection. In light of this, recent works (Guo et al., 2021b; Peng et al., 2022) start to explore the incorporation of point clouds in 3D object detection for performance improvement. One line of work (Wang et al., 2019b) projects each points to the image to form depth map labels, and subsequently trains a depth estimator to explicitly extract the spatial information. Such a paradigm generates intermediate products, *i.e.,* depth prediction maps, therefore introducing extra computational cost. Another line of work (Chong et al., 2021) is to leverage the teacher-student paradigm for

---
*Corresponding author

knowledge transfer. (Chong et al., 2021) projects the LiDAR points to the image plane, constructing a 2D input for the teacher model. Since the student and teacher models are exactly structurally identical, feature mimicking can be naturally conducted under the framework. Although it solves the alignment issue across different modalities, it misses the opportunity of pursuing a strong LiDAR-based teacher, which is indeed important in the knowledge distillation (KD) paradigm. Recently, UVTR (Li et al., 2022a) propose to distill the cross-modal knowledge in the voxel space, while maintaining the structure of respective detectors. However, it directly forces the 2D branch to imitate the 3D features, ignoring the divergence between different modalities.

In this work, by carefully examining the non-homogenous features represented in the 2D and 3D spaces, we explore the incorporation of knowledge distillation for multi-view 3D object detection. Yet, there are two technical challenges. First, the views of images and LiDAR points are different, *i.e.,* camera features are in the perspective view, while LiDAR features are in the 3D/bird's-eye view. Such a view discrepancy indicates that a natural one-to-one imitation (Romero et al., 2014) may not be suitable. Second, RGB images and point clouds hold respective representations in their own modalities. Therefore, it can be suboptimal to mimic features directly, which is commonly adopted in the 2D detection paradigm (Yang et al., 2022a; Zhang & Ma, 2020).

We address the above challenges by designing a cross-modal BEV knowledge distillation framework, namely **BEVDistill**. Instead of constructing a separate depth estimation network or explicitly projecting one view into the other one, we convert all features to the BEV space, maintaining both geometric structure and semantic information. Through the shared BEV representation, features from different modalities are naturally aligned without much information loss. After that, we adaptively transfer the spatial knowledge through both *dense* and *sparse* supervision: (i) we introduce soft foreground-guided distillation for non-homogenous dense feature imitation, and (ii) a sparse style instance-wise distillation paradigm is proposed to selectively supervise the student by maximizing the mutual information.

Experimental results on the competitive nuScenes dataset demonstrate the superiority and generalization of our BEVDistill. For instance, we achieve 3.4 NDS and 2.7 NDS improvements on a competitive multi-view 3D detector, BEVFormer (Li et al., 2022c), under the single-frame and multi-frame settings, respectively. Besides, we also present extensive experiments on lightweight backbones, as well as detailed ablation studies to validate the effectiveness of our method. Notably, our best model reaches 59.4 NDS on nuScenes test leaderboard, achieving new state-of-the-art results among all the published multi-view 3D detectors.

## 2 RELATED WORK

### 2.1 VISION-BASED 3D OBJECT DETECTION

Vision-based 3D object detection aims to detect object locations, scales, and rotations, which is of great importance in autonomous driving (Xie et al., 2022; Jiang et al., 2022; Zhang et al., 2022) and augmented reality (Azuma, 1997). One line of work is to detect 3D boxes directly from single images. Mono3D (Chen et al., 2016) utilizes traditional methods to lift 2D objects into 3D space with semantic and geometrical information. In the consideration that objects located at different distances appear on various scales, D4LCN (Ding et al., 2020) proposes to leverage depth prediction for convolutional kernel learning. Recently, FCOS3D (Wang et al., 2021a) extends the classical 2D paradigm FCOS (Tian et al., 2019b) into monocular 3D object detection. It transforms the regression targets into the image domain by predicting its 2D-based attributes. Further, PGD (Wang et al., 2022b) introduces relational graphs to improve the depth estimation for object localization. MonoFlex (Zhang et al., 2021) argues that objects located at different positions should not be treated equally and presents auto-adjusted supervision.

Another line of work predicts objects from multi-view images. DETR3D (Wang et al., 2022c) first incorporates DETR (Carion et al., 2020) into 3D detection by introducing a novel concept: 3D reference points. After that, Graph-DETR3D (Chen et al., 2022b) extends it by enriching the feature representations with dynamic graph feature aggregation. Different from the above methods, BEVDet (Huang et al., 2021) leverages the Lift-splat-shoot (Philion & Fidler, 2020) to explicitly project images into the BEV space, followed by a traditional 3D detection head. Inspired by the recently developed attention mechanism, BEVFormer (Li et al., 2022c) automates the cam2bev process with

a learnable attention manner and achieves superior performance. PolarFormer (Jiang et al., 2022) introduces polar coordinates into the model construction of BEV space and greatly improves the performance. Moreover, BEVDepth (Li et al., 2022b) improves BEVDet by explicitly supervising depth prediction with projected LiDAR points and achieves state-of-the-art performance.

## 2.2 KNOWLEDGE DISTILLATION IN OBJECT DETECTION

Most KD approaches for object detection focus on transferring knowledge among two homogenous detectors by forcing students' predictions to match those of the teacher (Chen et al., 2017; Dai et al., 2021; Zheng et al., 2022). More recent works (Guo et al., 2021a; Wang et al., 2019a) find that imitation of feature representations is more effective for detection. A vital challenge is to determine which feature regions should be distilled from the teacher model. FGFI (Wang et al., 2019a) chooses features that are covered by anchor boxes whose IoU with GT is larger than a certain threshold. PGD (Yang et al., 2022a) only focuses on a few key predictive regions using a combination of classification and regression scores as a measure of quality.

Despite a large number of works discussing the KD in object detection, only a few of them consider the multi-modal setting. MonoDistill (Chong et al., 2021) projects the points into the image plane and applies a modified image-based 3D detector as the teacher to distill knowledge. Such a paradigm solves the alignment issue naturally, however, it misses the chance to pursue a much stronger point-based teacher model. LIGA-stereo (Guo et al., 2021b) leverages the information of LiDAR by supervising the BEV representation of the vision-based model with the intermediate output from the SECOND (Yan et al., 2018). Recently, UVTR (Li et al., 2022a) presents a simple approach by directly regularizing the voxel representations between the student and teacher models. Both of them choose to mimic the feature representations across the models, while ignoring the divergence between different modalities.

## 3 METHODS

In this section, we introduce our proposed BEVDistill in detail. We first give an overview of the whole framework in Figure 1 and clarify the model designs of the teacher and student model in Section 3.1. After that, we present the cross-modal knowledge distillation approach in Section 3.2, which consists of two modules: dense feature distillation and sparse instance distillation.

## 3.1 BASELINE MODEL

**Student Model.** We adopt current state-of-the-art camera-based detector, BEVFormer (Li et al., 2022c), as our student model. It consists of an image backbone for feature extraction, a spatial cross-attention module for `cam2bev` view transformation, and a transformer head for 3D object detection. Besides, it provides a temporal cross-attention module to perceive subsequential multi-frame information for better predictions.

**Teacher Model.** In order to keep consistent with the student model, we select Object-DGCNN (Wang & Solomon, 2021) as our teacher model. For simplicity and generality, we replace the DGCNN attention with a vanilla multi-scale attention module. It first casts 3D points into the BEV plane and then conducts one-to-one supervision with transformer-based label assignment. We train the model by initializing it from a pre-trained CenterPoint model.

## 3.2 BEVDISTILL

BEVDistill follows the common knowledge distillation paradigm, with a 3D point cloud detector as teacher and an image detector as student. Different from previous knowledge distillation methods, which keep the same architecture for both student and teacher models (except for backbone), BEVDistill explores a more challenging setting with non-homogenous representations.

### 3.2.1 DENSE FEATURE DISTILLATION

In order to conduct dense supervision for feature distillation, we first need to determine the BEV features generated from both models. For the student, we directly adopt the BEV features maps

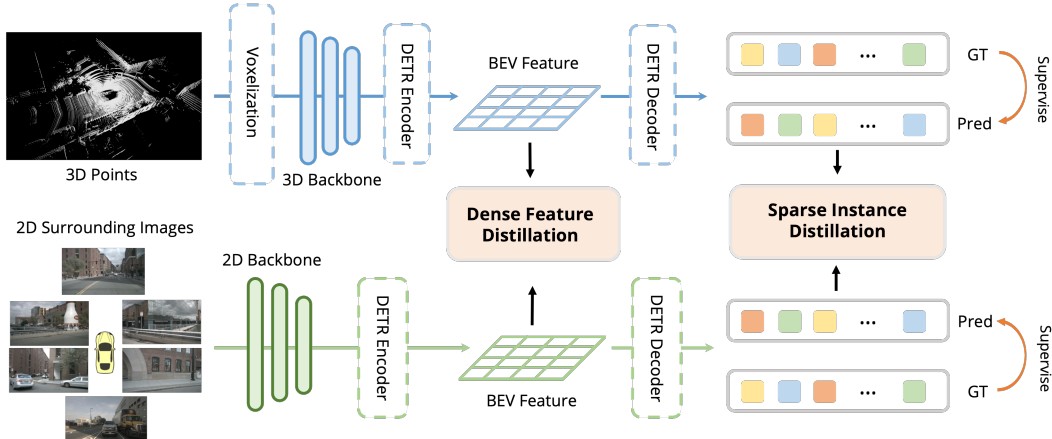

Figure 1: The overall architecture of the proposed BEVDistill consists of three main components: a transformer-based LiDAR detector, a transformer-based image detector, and the proposed cross-modal distillation modules. It follows the teacher-student paradigm with both RGB images and LiDAR points as the inputs and keeps the original structure of respective model structures.

$F^{2D}$ produced by the `BEVTransformerEncoder`. In order to align the feature representations between the teacher and student, we choose the same BEV feature $F^{3D}$ outputted by the transformer encoder for the teacher model. Different from LIGA-Stereo (Guo et al., 2021b) which directly mimics the BEV features extracted from the 3D backbone, we postpone such supervision after the transformer encoder (illustrated in Figure 1), which offers more opportunities for the network to align information across different modalities.

Previous works (Romero et al., 2014; Li et al., 2022a) directly force the student to mimic the feature map from the teacher and achieve significant performance improvement:

$$L_{feat} = \frac{1}{H \times W} \sum_{i}^{H} \sum_{j}^{W} ||F_{ij}^{3D} - F_{ij}^{2D}||_2, \qquad (1)$$

where $H, W$ denotes the width and height of the distilled feature map, and $|| \cdot ||_2$ is the $L_2$ norm. However, such a strategy may not work well under the cross-modal feature setting. Although the view discrepancy is eliminated by projecting both features into the BEV plane, there is still domain discrepancy between different modalities: *despite the same scenes being captured with the point clouds and images with careful alignment, the representation itself can be diverse under different modalities*. For instance, images holds pixels for both foreground and backgrounds regions, however LiDAR points only appears when there exists objects to reflect the ray.

Consider that the regional 3D features can hold meaningful information only if the point exists, we regularize the distillation within the foreground regions. Besides, we notice that the boundary of foreground could also provides useful information. Therefore, instead of performing hard supervision on each foreground region, we introduce a soft-supervision manner similar to that in (Zhou et al., 2019). Concretely, we draw Gaussian distribution for each ground truth center $(x_i, y_i)$ in the BEV space with,

$$w_{i,x,y} = \exp(-\frac{(x_i - \hat{x}_i)^2 + (y_i - \hat{y}_i)^2}{2\sigma_i^2}), \qquad (2)$$

where $\sigma_i$ is a constant (set to 2 by default), indicating the object size standard deviation. Since the feature maps are class-agnostic, we merge all $w_{i,x,y}$ into one single mask $W$. For overlapping regions among different $w_{i,x,y}$ at the same position, we simply take the maximum value of them.

After that, we enforce the student to mimic the feature with the foreground-guided mask $W$ for dense feature distillation:

$$L_{feat} = \frac{1}{H \times W \times \sum max(W_{i,j})} \sum_i^H \sum_j^W max(W_{ij}) ||F_{ij}^{3D} - F_{ij}^{2D}||_2. \tag{3}$$

This foreground-guided re-weight strategy allows the model to focus on the foreground regions from the teacher, and at the same time avoid imitation of useless empty 3D features in the background regions. Such a manner is counter to the finding in previous homogeneous knowledge distillation works (Yang et al., 2022c).

### 3.2.2 SPARSE INSTANCE DISTILLATION

Instance-level distillation can be easily conducted under the dense prediction setting, where only a pixel-to-pixel mapping is needed for supervision. However, both the student and teacher models in BEVDistill act in a sparse-prediction style. Therefore, a set-to-set mapping is required to ensure instance-level distillation. To achieve the goal, we simply follow the practice in (Wang & Solomon, 2021) to construct the correspondence between the student and teacher predictions. Specifically, suppose the categorical and localization predictions output by the $i_{th}$ query from the teacher model are $c_i^T$ and $b_i^T$, and the student can be represented as $c_i^S$ and $b_i^S$, a permutation of $\hat{\sigma}$ can be found between the output set of the teacher and the student with the lowest cost:

$$\hat{\sigma} = \arg\min \sum_i^N \mathcal{L}_{match}(y_i, \hat{y}_i), \tag{4}$$

where $\mathcal{L}_{match}(y_i, \hat{y}_i)$ is a pair-wise matching cost, which is defined as:

$$\mathcal{L}_{match}(y_i, \hat{y}_i) = -\log c_{\sigma(i)}^S(c_i^T) + ||b_i^T, b_{\sigma(i)}^S||_1. \tag{5}$$

However, we empirically find such a vanilla distillation approach works poorly in a cross-domain supervision manner. There are two main issues that impede the model from further performance improvement. On one hand, not all predictions from the teacher should be treated equally as valuable cues, since most of the predictions are false positives with a low classification score. On the other hand, though the classification logits can represent rich knowledge (Hinton et al., 2015), it may not hold when the input data are different. Directly distilling these prediction will introduce great noise to the model and deteriorate the performance. To tackle the problem, we exploit reliable quality score derived from the teacher model to measure the importance of the instance-level pseudo labels and use the score as soft re-weighted factors to eliminate the potential noise predictions made by the teacher. Specifically, we consider the classification score $c_i$ (categorical information) and the IoU between the prediction $b_i^{pred}$ and the ground truth $b_i^{GT}$ (localization information) together to formulate the quality-score $q_i$:

$$q_i = (c_i)^\gamma \cdot \text{IoU}(b_i^{GT}, b_i^{pred})^{1-\gamma}. \tag{6}$$

The quality score serves as an indicator to guide the student on which teacher's predictions should be paid more weight. Therefore, the final instance-level distillation can be written as:

$$\mathcal{L}_{inst} = \sum_i^N -q_i \left(\alpha \mathcal{L}_{cls}(c_{\sigma(i)}^S, c_i^T) + \beta \mathcal{L}_{box}(b_i^T, b_{\sigma(i)}^S)\right), \tag{7}$$

where $\mathcal{L}_{cls}$ is the KL divergence loss, $\mathcal{L}_{box}$ is the $L_1$ loss in (Wang & Solomon, 2021), and $\alpha$, $\beta$ are reweighted factors to balance the supervision (set to 1.0 and 0.25 by default). Given that the prediction probability mass function over classes contains richer information than one-hot labels, student models are proven to be benefited from such extra supervision (Hinton et al., 2015). However, this low dimensional prediction distribution (number of classes) means that only a few amount of knowledge is encoded, thus limiting the knowledge that can be transferred, especially under the cross-modal setting. Besides, the representational knowledge is often structured, with implicit complex interdependencies across different dimensions, while the KL objective treats all dimensions

independently. To overcome the problem, instead of minimizing the KL divergence over two distributions, we choose to directly maximize the mutual information (MI) between the representations $h^S, h^T$ at the penultimate layer (before logits) from the teacher and student networks:

$$I(h^S; h^T) = \text{KL}(p(h^T, h^S) \| \mu(h^T) v(h^S)), \tag{8}$$

where $p(h^T, h^S)$ is the joint distribution and the marginal distributions $\mu(h^T), v(h^S)$. We then define a distribution $q$ conditioned on $\eta$ that captures whether the pair is congruent ($q(\eta = 1)$) or incongruent ($q(\eta = 0)$):

$$q(h^T, h^S | \eta = 1) = p(h^T, h^S), q(h^T, h^S | \eta = 0) = \mu(h^T) v(h^S). \tag{9}$$

With Bayes' rule, we can obtain the posterior for $\eta = 1$:

$$q(\eta = 1 | h^T, h^S) = \frac{p(h^T, h^S)}{p(h^T, h^S) + \mu(h^T) v(h^S)}, \tag{10}$$

By taking the logarithm of both sides, we have

$$\log q(\eta = 1 | h^T, h^S) \leq \log \frac{p(h^T, h^S)}{p(h^T, h^S) + \mu(h^T) v(h^S)} = I(h^T, h^S). \tag{11}$$

Therefore, the objective can be transformed to maximize the lower bound of MI. Since there is no closed form for $q(\eta = 1 | h^T, h^S)$, we utilize a neural network $g$, called critic (Tian et al., 2019a) to approximate with NCE loss (see Appendix A for detailed implementation):

$$L_{cls}(h^T, h^S) = E_{q(h^T, h^S | \eta=1)}[\log g(h^T, h^S)] + E_{q(h^T, h^S | \eta=0)}[\log (1 - g(h^T, h^S))]. \tag{12}$$

To this end, the final sparse instance distillation loss can be represented as:

$$\mathcal{L}_{inst} = \sum_i^N -q_i \left( \alpha \mathcal{L}_{cls}(h_i^S, h_i^T) + \beta \mathcal{L}_{box}(b_i^T, b_{\sigma(i)}^S) \right). \tag{13}$$

## 4 EXPERIMENTS

### 4.1 DATASET AND EVALUATION METRICS

We conduct the experiments on the NuScenes dataset (Caesar et al., 2020), which is one of the most popular datasets for 3D object detection. It consists of 700 scenes for training, 150 scenes for validation, and 150 scenes for testing. For each scene, it includes 6 camera images (front, front left, front right, back left, back right, back) to cover the whole viewpoint, and a 360° LiDAR point cloud. Camera matrixes including both intrinsic and extrinsic are provided, which establish a one-to-one correspondence between each 3D point and the 2D image plane. There are 23 categories in total, and only 10 classes are utilized to compute the final metrics according to the official evaluation code.

We adopt the official evaluation toolbox provided by nuScenes. We report nuScenes Detection Score (NDS) and mean Average Precision (mAP), along with mean Average Translation Error (mATE), mean Average Scale Error (mASE), mean Average Orientation Error (mAOE), mean Average Velocity Error (mAVE), mean Average Attribute Error (mAAE) in our experiments. The NDS score is a weighted sum of mean Average Precision (mAP) and the aforementioned True Positive (TP) metrics, which is defined as NDS $= \frac{1}{10}[5 \times mAP + \sum_{\text{mTP} \in TP} (1 - \min(1, \text{mTP}))]$.

### 4.2 IMPLEMENTATION DETAILS

Our codebase is built on MMDetection3D (Contributors, 2020) toolkit. All models are trained on 8 NVIDIA A100 GPUs. We first train the 3D detector with a voxel size of $(0.1m, 0.1m, 0.2m)$ followed in (Yin et al., 2021). During the distillation phase, the batch size is set to 1 per GPU with an initial learning rate of 2e-4. Unless otherwise specified, we train the models for $2 \times$ schedule (24 epochs) with a cyclic policy. The input image size is set to $1600 \times 900$ and the grid size of the BEV plane in BEVFormer is set to $128 \times 128$. FP16 training is enabled for GPU memory management.

Table 1: Comparison of recent distillation works on the nuScenes validation set with ResNet-50 image backbone. ∗ indicates our own re-implementation with modifications.

| Detector | Setting | NDS↑ | mAP↑ |
|---|---|---|---|
| BEVFormer | Teacher | 67.4 | 61.5 |
| | Student | 42.3 | 35.2 |
| | FitNet (Romero et al., 2014) | 41.1 | 34.4 |
| | Set2Set (Wang & Solomon, 2021) | 41.0 | 33.1 |
| | MonoDistill∗ (Chong et al., 2021) | 42.9 | 36.4 |
| | UVTR (Li et al., 2022a) | 43.1 | 36.2 |
| | BEVDistill (**Ours**) | **45.7**(+3.4) | **38.6**(+3.4) |
| BEVFormer-T | Teacher | 67.4 | 61.5 |
| | Student | 48.8 | 38.3 |
| | FitNet (Romero et al., 2014) | 48.0 | 37.3 |
| | Set2Set (Wang & Solomon, 2021) | 47.9 | 37.5 |
| | MonoDistill∗ (Chong et al., 2021) | 49.5 | 39.0 |
| | UVTR (Li et al., 2022a) | 50.1 | 39.4 |
| | BEVDistill (**Ours**) | **51.5**(+2.7) | **40.7**(+2.4) |

## 4.3 MAIN RESULTS

We first implement BEVFormer and BEVFormer-T on the nuScenes validation subset with ResNet-50 backbone and compare BEVDistill with other knowledge distillation methods. The final results are shown in Table 1. Our BEVDistill greatly boosts its vanilla student model, BEVFormer, by more than 3.4/2.3 mAP and 3.4/2.7 NDS, respectively, without introducing any extra inference cost. Classical and effective distillation methods like FitNet (Romero et al., 2014), fail to bring any enhancement and even deteriorate the final performance, *i.e.,* -1.2% NDS. The Set2Set (Wang & Solomon, 2021) also gets struck in the detection accuracy (-1.3% NDS) due to the miss consideration of the divergence between different modality features. UVTR (Li et al., 2022a), which is the first work to leverage 3D points for multi-view 3D object detection, manages to get an 0.8 NDS improvement. However, it only focuses on feature-based distillation while missing the potential benefits in other aspects. We also reimplement MonoDistill (Chong et al., 2021) by extending its monocular version to a multi-view one, yielding an improvement of 0.7 NDS.

## 4.4 COMPARISON WITH STATE-OF-THE-ARTS

In addition to offline results, we also report the detection performance on nuScenes test leaderboard compared to various multi-view 3D detection approaches to further explore the potential of BEVDistill. The results are shown in Table 2. Our final model is based on current state-of-the-art model BEVDepth with an input image size of 640 × 1600. We adopt ConvNeXt-base (Liu et al., 2022d) as our image encoder with a BEV space of 256 × 256. The model is trained 20 epochs with CBGS and no test-time augmentation is applied. It surpasses all other detectors, including the recently developed PETRv2, BEVFormer, and BEVDet4D by more than 1.0 NDS, achieving new state-of-the-art on this competitive benchmark.

## 4.5 ABLATION STUDIES

In this section, we provide detailed ablation studies on different components of our model. For efficiency, we adopt BEVFormer-R50 as base model with 1× schedule on 1/2 training subset.

### 4.5.1 MAIN ABLATIONS

To understand how each module contributed to the final detection performance in BEVDistill, we test each component independently and report its performance in Table 3. The overall baseline starts from 42.2 NDS. When dense feature distillation is applied, the NDS is raised by 2.5 points, which

Table 2: Comparison of recent works on the nuScenes testing set. "L" and "C" indicate LiDAR and Camera, respectively. ‡ denotes our reimplementation borrowed from BEVDet.

| Method | M. | NDS | mAP | mATE | mASE | mAOE | mAVE | mAAE | Reference |
|--------|-----|------|------|-------|-------|-------|-------|-------|-----------|
| PointPillar | L | 45.3 | 30.5 | - | - | - | - | - | CVPR2019 |
| CenterPoint | L | 65.5 | 58.0 | - | - | - | - | - | CVPR2021 |
| BEVFusion | L+C | 72.9 | 70.2 | 0.239 | 0.329 | 0.260 | 0.134 | 0.153 | Arxiv2022c |
| FCOS3D | C | 42.8 | 35.8 | 0.690 | 0.249 | 0.452 | 1.434 | **0.124** | ICCVW2021a |
| Ego3RT | C | 47.3 | 42.5 | 0.549 | 0.264 | 0.433 | 1.014 | 0.145 | ECCV2022 |
| DETR3D | C | 47.9 | 41.2 | 0.641 | 0.255 | 0.394 | 0.845 | 0.133 | CoRL2022c |
| BEVDet | C | 48.8 | 42.4 | 0.524 | 0.242 | 0.373 | 0.950 | 0.148 | Arxiv2021 |
| G-DETR3D | C | 49.5 | 42.5 | 0.621 | 0.251 | 0.386 | 0.790 | 0.128 | MM2022b |
| PETR | C | 50.4 | 44.1 | 0.593 | 0.249 | 0.384 | 0.808 | 0.132 | ECCV2022a |
| UVTR | C | 55.1 | 47.2 | 0.577 | 0.253 | 0.391 | 0.508 | 0.123 | NeurIPS2022a |
| BEVDet4D | C | 56.9 | 45.1 | 0.511 | 0.241 | 0.386 | 0.301 | 0.121 | Arxiv2022 |
| BEVFormer | C | 56.9 | 48.1 | 0.582 | 0.256 | 0.375 | 0.378 | 0.126 | ECCV2022c |
| PETRv2 | C | 58.2 | 49.0 | 0.561 | 0.243 | 0.361 | 0.343 | 0.120 | Arxiv2022b |
| BEVDepth ‡ | C | 58.9 | 49.1 | 0.484 | 0.245 | 0.377 | 0.320 | 0.132 | Arxiv2022b |
| + BEVDistill | C | **59.4** | **49.8** | 0.472 | 0.247 | 0.378 | 0.326 | 0.125 | - |

Table 3: Effectiveness of each component in BEVDistill. Results are reported on BEVFormer-R50 with standard 2 × schedule setting.

| | Dense Feature Distill | Sparse Instance Distill | NDS↑ | mAP↑ | mAOE↓ |
|---|----|----|------|------|-------|
| a | | | 42.3 | 35.2 | 0.428 |
| b | ✓ | | 44.7 | 38.0 | 0.416 |
| c | | ✓ | 43.9 | 37.1 | 0.410 |
| d | ✓ | ✓ | **45.7** | **38.6** | **0.399** |

validates the finding that disentangling the foreground regions with soft supervision can greatly promote the student's detection ability. Then, we add the sparse instance distillation, which brings us a 1.7% NDS enhancement. Finally, the NDS score achieves 45.7 when all components are applied, yielding a 3.4 % absolute improvement, validating the effectiveness of BEVDistill.

### 4.5.2 Ablations on Dense Feature Distillation

**Comparison with other feature-based distillation mechanisms.**

Feature-based distillation methods have been extensively explored in both image classification (Hinton et al., 2015) and object detection tasks (Yang et al., 2022b;c). Therefore, we compare our dense feature distillation module with other competitive feature-based distillation approaches. The results are shown in Table 4. FitNet distills all pixels across 2D and 3D features with equal supervision, yielding little improvement. FGD (Yang et al., 2022b) is current state-of-the-art knowledge distillation approach in 2D object detection. However, it neglects the divergence between different modalities across features, therefore gets the worse performance. UVTR (Li et al., 2022a) also mimics the features directly but it only selects positive regions (query points) for feature imitation. Compared with the above methods, BEVDistill considers both modality divergence and region importance for feature imitation, providing the highest accuracy.

**Strategies on foreground mask generation.**

As mentioned in Section 3.2.1, not all positions in BEV feature representation should be treated equally. How to select the most informative regions for feature imitation is also non-trivial. Hence, we provide detailed experiments on how to generate the foreground imitation mask for distillation. Intuitively, we can directly select the center points of the ground truth objects (GT-Center) or the

Table 4: Comparison on feature-based knowledge distillation approaches.

| Method | NDS | mAP | mAOE |
|--------|-----|-----|------|
| - | 37.5 | 33.3 | 0.515 |
| FitNet | 37.8 | 34.4 | 0.549 |
| FGD | 36.5 | 32.9 | 0.534 |
| UVTR | 38.1 | 34.5 | 0.513 |
| BEVDistill | **38.9** | **34.9** | **0.501** |

Table 5: Comparison on foreground mask generation approaches.

| Method | NDS | mAP | mAOE |
|--------|-----|-----|------|
| - | 37.5 | 33.3 | 0.515 |
| GT-Center | 37.7 | 34.2 | 0.546 |
| Query-Center | 38.1 | 34.4 | 0.513 |
| Pred-Heatmap | 38.6 | 34.5 | 0.533 |
| GT-Heatmap | **38.9** | **34.9** | **0.501** |

query reference points (Query-Center) as imitation instances. However, such a strategy directly abandons most areas, which may contain useful information. Therefore, we attempt to convert such a 'hard' decision into the 'soft' manner with a heatmap mask, which can be the model classification prediction from the teacher (Pred-Heatmap) or the ground truth heatmap (GT-Heatmap). The final results are shown in Table 5. The soft foreground mask generation is much better compared to the hard one. We choose the GT-Heatmap strategy in our experiments due to its highest score.

### 4.5.3 ABLATIONS ON SPARSE INSTANCE DISTILLATION

Table 6: Ablations on loss selection for classification and regression branches.

| cls | reg | NDS | mAP | mAOE |
|-----|-----|-----|-----|------|
| CE | - | 35.8 | 31.7 | 0.544 |
| - | L1 | 38.1 | 33.9 | 0.521 |
| CE | L1 | 37.0 | 32.7 | 0.531 |
| InfoNCE | L1 | **39.1** | **34.6** | **0.474** |

Table 7: Ablations on critic $g$ selection and pair generation.

| critic $g$ | Pair | NDS | mAP |
|------------|------|-----|-----|
| - | - | 37.5 | 33.3 |
| NCS Loss | Pos | 38.0 | 33.7 |
| CE Loss | Pos | 35.8 | 31.7 |
| InfoNCE Loss | Pos + Neg | 38.6 | 34.5 |

**Ablations on the instance distillation loss.**

Instance-level distillation is a necessary module in detection-based knowledge distillation. However, such a component has been seldom discussed in a transformer-based sparse prediction setting. In this ablation, we conduct detailed experiments on the loss selection in Table 6. We first conduct the commonly adopted cross-entropy and L1 loss for the classification and regression branches. The result is surprisingly lower than its vanilla baseline. When adding them respectively, we figure out that the cross-entropy is the key issue leading to performance degradation. To avoid the problem, we replace it with the InfoNCE loss along with the contrastive paradigm, and an immediate improvement is observed, which validates our assumption in Section 3.2.2.

**Strategies on selection of critic $g$.**

Since most self-supervised learning approaches are based on homogenous representations, exploring the suitable contrastive paradigm for cross-modality is important. Note that features located in the same positions in the 2D and 3D BEV feature maps are considered positive pairs while the rest are negative pairs. We compare three prototypes of critic $g$ and report the results in Table 7. We first adopt the classic InfoNCE loss, which provides the most improvement. We also conduct experiments on positive pairs only with NCS (negative cosine similarity) and CE (cross-entropy) loss, while gets little enhancement, validating the proof in Section 3.2.2.

## 5 CONCLUSION

In this paper, we introduce a cross-modal knowledge distillation framework for multi-view 3D object detection, namely BEVDistill. It unifies different modalities in the BEV space to ease the modal alignment and feature imitation. With the dense feature distillation module, it can effectively transfer rich teacher knowledge to the student across 2D and 3D data. BEVDistill also explores a sparse instance distillation approach under the transformer-based detection paradigm. Without bells and whistles, it achieves new state-of-the-art performance on the competitive nuScenes test leaderboard.

ACKNOWLEDGMENTS

This work was supported by the JKW Research Funds under Grant 20-163-14-LZ-001-004-01, and the Anhui Provincial Natural Science Foundation under Grant 2108085UD12. We acknowledge the support of GPU cluster built by MCC Lab of Information Science and Technology Institution, USTC.

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

## A    IMPLEMENTATION OF CRITIC $g$

Firstly, we adopt bilinear $\mathrm{F}_{bilinear}$ to obtain the paired 2D/3D features $f_i^{2D}, f_i^{3D}$ on student and teacher BEV features maps $F^{2D}, F^{3D}$ according to the reference location $r_i$ predicted by each student query $q_i$:

$$f_i^{2D} = \mathrm{F}_{bilinear}(F^{2D}, r_i), \quad f_i^{3D} = \mathrm{F}_{bilinear}(F^{3D}, r_i). \tag{14}$$

Note that we adopt query points from 6 stages which actually form $900 \times 6$ feature pairs. Each 2D and 3D feature located at the same position are viewed as a positive pair. We use separate projection heads for 2D and 3D features since they originated from different modalities. Each projection head is a 3-layer MLP ($g^{2D}, g^{3D}$) and the final loss can be written as

$$
\begin{aligned}
x_i^{2D} &= g^{2D}(f_i^{2D}), \quad x_i^{3D} = g^{3D}(f_i^{3D}) \\
L &= -\log \frac{\exp(x_i^{2D} \cdot x_i^{3D}/\tau)}{\sum_{j=0, j \neq i}^{K} \exp(f_i^{2D} \cdot f_j^{3D}/\tau)}
\end{aligned}
\tag{15}
$$

where $\tau$ is the temperature. Different from the original implementation in (He et al., 2020), we do not collate samples from other cards during distributed training since there are enough negative samples (5400-1=5399) for each card.

## B    MORE EXPERIMENTAL RESULTS

### B.1    EXPERIMENTS ON BEVFORMER WITH LARGER BACKBONES

Instead of the results conducted on ResNet-50 backbone, we also provide detailed experiments with stronger backbones, ResNet-101. We keep all experimental settings the same in Section 4.5 except for the image backbone and the results are shown in Table 8. From the table, we can conclude that BEVDistill can still consistently improve the competitive baseline BEVFormer, even with a much stronger ResNet-101 backbone by more than 2.3 and 1.8 NDS.

Table 8: Experimental results of BEVDistill with BEVFormer on ResNet-101 DCN backbone, under both single-frame and multi-frame settings.

| Detector | Setting | NDS↑ | mAP↑ |
|---|---|---|---|
| BEVFormer | Teacher | 67.4 | 61.5 |
| | Student | 44.5 | 37.4 |
| | +BEVDistill | **46.8**(+2.3) | **38.9**(+1.5) |
| BEVFormer-T | Teacher | 67.4 | 61.5 |
| | Student | 50.6 | 40.5 |
| | +BEVDistill | **52.4**(+1.8) | **41.7**(+1.2) |

### B.2    DISTILLING TO A LIGHTWEIGHT BACKBONE

Knowledge distillation is actually one of the most common solutions to model compression. It is usually adopted to transfer useful information from a large stronger model to a lightweight weaker model, which is suitable for deployment on the edge. Considering that vision-based detectors are fed with a batch of images for surrounding view inputs, it is extremely important to reduce the size of the image backbone. With this in mind, we conduct experiments with smaller backbones, including ResNet-18 and MobileNetV2, which is a more demanded setting in real applications. The final results are shown in Table 9. BEVDistill improves the ResNet-18 and MobileNetV2 backbones by more than 2.1/1.7 NDS, pointing its great potential for resource-limited applications including autonomous driving and edge computing.

Table 9: Experimental results of BEVDistill on smaller image backbones: ResNet-18 and Mo-
bileNetV2 on nuScenes validation subset.

| Detector | Backbone | Setting | NDS↑ | mAP↑ |
|---|---|---|---|---|
| BEVFormer | ResNet-18 | Teacher | 67.4 | 61.5 |
| | | Student | 37.3 | 30.0 |
| | | + BEVDistill | **40.2**(+2.9) | **32.0**(+2.0) |
| BEVFormer-T | ResNet-18 | Teacher | 67.4 | 61.5 |
| | | Student | 43.7 | 32.1 |
| | | + BEVDistill | **45.8**(+2.1) | **33.9**(+1.8) |
| BEVFormer | MobileNet-V2 | Teacher | 67.4 | 61.5 |
| | | Student | 37.0 | 29.3 |
| | | + BEVDistill | **38.8**(+1.8) | **30.8**(+1.5) |
| BEVFormer-T | MobileNet-V2 | Teacher | 67.4 | 61.5 |
| | | Student | 43.1 | 32.0 |
| | | + BEVDistill | **44.8**(+1.7) | **33.5**(+1.5) |

Table 10: Experimental results of BEVDistill with BEVDepth on ResNet-50 backbone.

| Detector | Setting | NDS↑ | mAP↑ |
|---|---|---|---|
| BEVDepth | Teacher | 66.4 | 60.3 |
| | Student | 44.0 | 31.7 |
| | +BEVDistill | **45.2**(+1.2) | **33.0**(+1.3) |

## B.3 EXPERIMENTS ON BEVDEPTH

To fully exploit the capability of this simple framework, we extend it to adapt the current state-
of-the-art multi-view 3D detector, BEVDepth. Since BEVDepth and BEVFormer both share BEV
space as the intermediate feature representation, we can directly apply dense feature distillation. As
for the instance distillation, BEVDepth conducts dense prediction similar to CenterPoint. Therefore,
we directly choose CenterPoint as our teacher model and only conduct instance distillation on the
foreground regions (*i.e.,* the center of ground truth objects). We conduct experiments on ResNet-50
backbone on nuScenes 1/2 training subset and report the results on validation subset in Table 10.
Surprisingly, BEVDistill can still improve its vanilla baseline by 1.2 NDS, even though BEVDepth
has already leveraged the information from LiDAR data with explicit depth supervision.

Table 11: Experimental results of BEVDistill with MV-FCOS3D++ and MV-FCOS4D++ on Waymo
Open Dataset. Following the practice in (Wang et al., 2022a), we sample every 5 frame to form the
training set.

| Detector | Backbone | BEVDistill | mAPL↑ | mAP↑ | mAPH↑ |
|---|---|---|---|---|---|
| MV-FCOS3D++ | ResNet-50 | | 32.6 | 45.1 | 42.7 |
| | ResNet-50 | ✓ | 34.3 | 46.8 | 44.4 |
| MV-FCOS4D++ | ResNet-50 | | 34.0 | 46.5 | 43.8 |
| | ResNet-50 | ✓ | 35.1 | 47.5 | 44.9 |

## B.4 EXPERIMENTS ON WAYMO OPEN DATASET

In this section, we report BEVDistill on another competitive public dataset, Waymo Open Dataset
(Sun et al., 2020) to further demonstrate the generalization of our approach. Due to the characteris-
tics of the camera-based detections, instead of adopting original mAP metrics, we choose the official

released LET-mAP (Hung et al., 2022) to evaluate the model performance. Since most multi-view 3D detectors do not report their performances on Waymo dataset, we select another competitive multi-view 3D detector, MV-FCOS3D++ (Wang et al., 2022a), which achieves $2^{rd}$ place at the Waymo Open Dataset Challenge–Camera-Only Track, as our baseline. The follow the same experimental settings in the official MV-FCOS3D++ repository. We select our teacher model as SECOND to adapt with the `Anchor3DHead` used by MV-FCOS3D++. The final results are reported in Table 11. It can be concluded that BEVDistill can consistently improves MV-FCOS3D++ and MV-FCOS4D++ (multi-frames input version of MV-FCOS3D++) by 1.7/1.1 mAPL, respectively, indicating the robustness and general of the proposed method.

## C   DISCUSSIONS

### C.1   WHY KL DIVERGENCE DISTILLATION ON SPARSE QUERY LOGITS FAILS TO IMPROVE?

KL logit distillation has been proven to be effective in plenty of works (Tian et al., 2019a; Hinton et al., 2015). However, it fails to yield improvements under the cross-modal distillation. By visualizing the training data and the predicted logits, we attribute one reason to this problem: for most previous distillations, both teacher and student are fed with the data in the same modality. Therefore, both of them should behave in a similar manner in every region. Such a hypothesis may not hold under different modalities: due to the characteristics of the modal sensor, the information collected by each sensor can be diverse. For instance, even if one person is occluded by another one in the image view, he may still be perceived by the LiDAR sensors. Besides, some instances can be easily detected by one modal sensor, while it may be hard on the other one, leading to different logits distribution predicted by the model. Directly forcing the student to mimic such distribution can deteriorate the inner knowledge learned by the student itself. Different from logit imitation, the contrastive paradigm only enforces the contexts encoded in each modality are the same, providing the freedom for the model to learn the representation encoded in the respective modality.

### C.2   WHAT DOES THE MODEL LEARN FROM ITS CROSS-MODAL TEACHER?

Different from the previous knowledge distillation framework, BEVDistill leverages LiDAR models to transfer knowledge. A natural question is: what does a multi-view vision-based student learn from its LiDAR-based 3D teacher? Actually, it is hard to figure out the explicit knowledge learned by the student, however, we can get some hints to some extent by comparing the detection performance evaluated from various aspects. The detailed results are shown in Table 12.

Table 12: Detailed metrics comparison between BEVFormer and BEVDistill on nuScenes validation dataset with ResNet-50 image backbone.

| Method | NDS | mAP | mATE | mASE | mAOE | mAVE | mAAE |
|---|---|---|---|---|---|---|---|
| BEVFormer | 42.3 | 35.2 | 0.755 | 0.271 | 0.428 | 0.870 | 0.208 |
| + BEVDistill | 45.7 | 38.6 | 0.693 | 0.264 | 0.399 | 0.802 | 0.199 |
| BEVFormer-T | 48.8 | 38.3 | 0.707 | 0.281 | 0.435 | 0.414 | 0.194 |
| + BEVDistill | 51.5 | 40.7 | 0.663 | 0.268 | 0.393 | 0.374 | 0.184 |

From the table we can find that the most significant improvements are mainly related to object localization, orientation and velocity estimation. LiDAR point clouds provide accurate localization information, which is of great benefit to localizing the object, including the absolute 3D space and its orientation. Both of them are more challenging to acquire from the camera view only. Note that the velocity estimation gets a 10% enhancement since the LiDAR model introduces more spatial information (*i.e.,* the LiDAR model takes 10 consecutive sweeps of point clouds as inputs instead of a single frame). As for the attribute prediction, which is mostly bound up with the classification tasks, BEVDistill fails to gain satisfying performance. We attribute the reason to the deficiency of point clouds in semantic representation compared to the imagery data, which may also account for the failure in KL distillation on the classification branch discussed in Section 3.2.2.

# D    QUALITATIVE RESULTS

We provide some qualitative results in Figure 2. More visualizations can refer to the supplementary materials, which are presented in a video format.

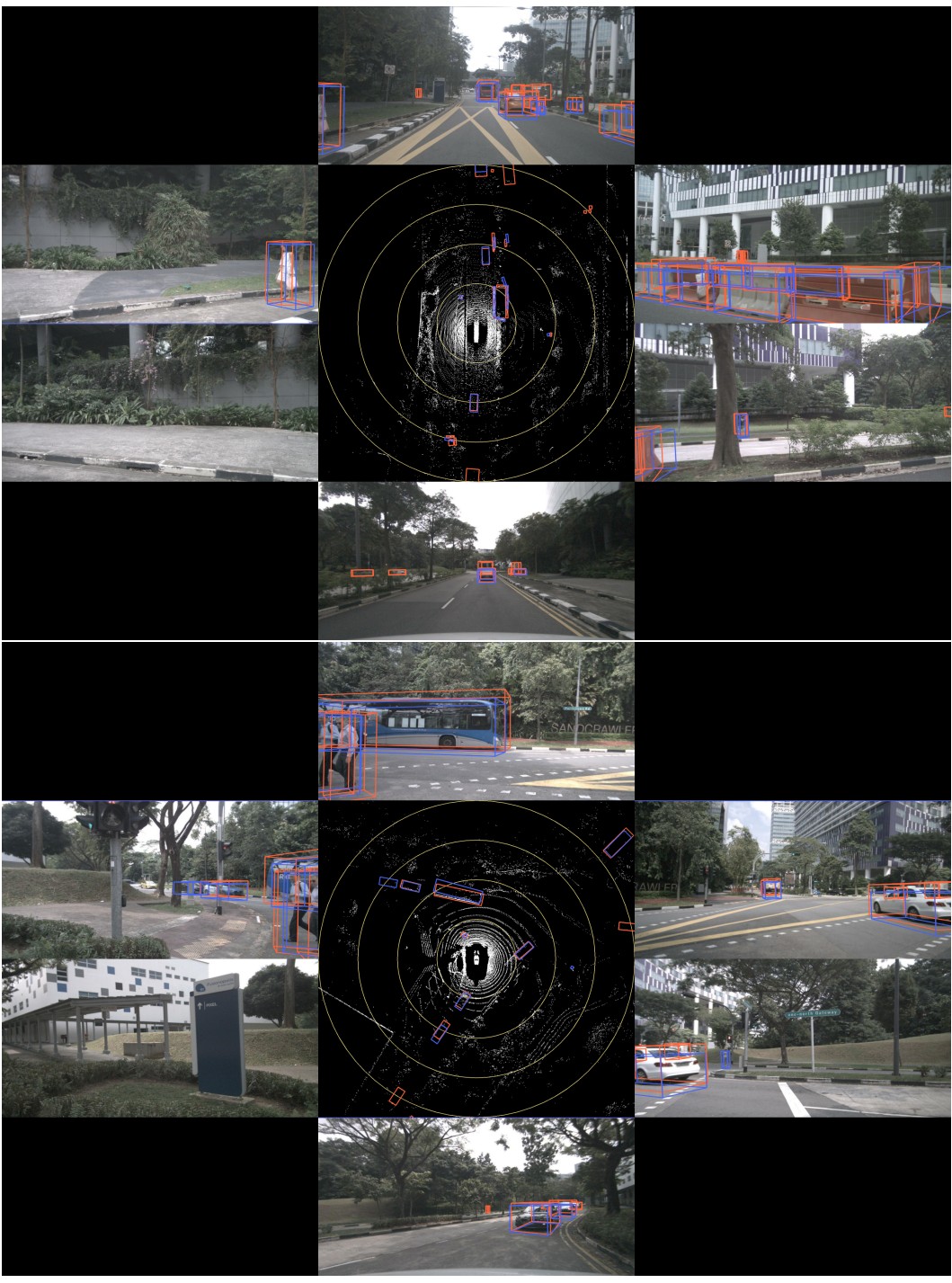

Figure 2: Visualization of the detection results predicted by BEVDistill. Best view in color: the prediction and ground truth are in blue and orange, respectively.

