# OpenReview forum: "BEVDistill: Cross-Modal BEV Distillation for Multi-View 3D Object Detection"
_ICLR.cc/2023/Conference — ICLR 2023 poster_

### Official Review · Reviewer_diMS · 2022-10-21

**Confidence:** 3
**Correctness:** 4
**Technical Novelty And Significance:** 2
**Empirical Novelty And Significance:** 2
**Recommendation:** 6

**Clarity, Quality, Novelty And Reproducibility:**

The overall writing and clear and easy to follow. But the novelty could be limited.

**Strength And Weaknesses:**

Strength:
---------------
1. This work focus on cross-modality distillation. It's reasonable to transfer geometry-aware knowledge to images during training.
2. The experiments are sufficient and proven to be effective on the nuScenes dataset.
3. The overall writing is clear and easy to follow.

Weakness:
---------------
1. Compared with previous work that uses feature distillation like LIGA-stereo and UVTR, the main contribution of this work is the sparse instance distillation. However, target-level distillation is not a novel thing in the distillation domain. And it only contributes 0.6% mAP gain and 1.0% NDS gain on top of the dense feature distill baseline in Table 3. That means most of the improvements are still from the dense feature distillation. This could make the contribution of this work limited.
2. The author claims "we postpone such supervision after the transformer encoder, which offers more opportunities for the network to align information across different modalities." in Section 3.2.1. But according to Figure 1, the dense feature distillation is still operated in the BEV representation. The reviewer is confusing the difference between it and UVTR, and why it offers more opportunities for the network to align information. Maybe the author can provide more details on the dense feature distillation.
3. Why the BEV feature in Figure 1 is 3D BEV? Is the height axis preserved here? It's better to add some illustrations in Section 3.2.1.
4. Writing: Section 4.2 can be moved to supplementary material. And it's better to report Table 10 in the main paper.

**Summary Of The Paper:**

This work proposes a cross-modality knowledge distillation framework for multi-view 3D object detection, called BEVDistill. It introduces two types of feature distillation, i.e., dense feature distillation and sparse instance distillation. Experiments prove the effectiveness of the proposed modules on the nuScenes dataset.

**Summary Of The Review:**

My main concern is the contribution of this work. It could be limited compared with previous cross-modality distillation methods. Please refer to the Weakness section for more details.

---

> ### Author Response · Authors · 2022-11-13
> **Response to Reviewer diMS**
>
> We sincerely appreciate your constructive and detailed comments.
>
> Q1: Feature distillation is also discussed in LIGA-Stereo and UVTR and the improvement of sparse instance distillation is not significant.
>
> For the dense BEV feature distillation, our approach differs from the LIGA-Stereo and UVTR, where we introduce a foreground-guided feature selection mask to overcome the problem caused by non-homogenous representations. As shown in Table 3 in our paper, the dense feature distillation module outperforms UVTR by 2.6/1.4 NDS, respectively, which is based on our key finding: *despite the same scenes being captured with the point clouds and images with careful alignment, the representation itself can be diverse under different modalities*. As for the sparse instance distillation module, the improvement on vanilla BEVFormer is 1.6 NDS, where the commonly utilized cross-entropy is -0.5 NDS, proving the superiority of our approach. This is based on our finding that representational knowledge is often structured, with implicit complex interdependencies across different dimensions, while the KL objective treats all dimensions independently, which is seldomly discussed in other knowledge distillation works. Overall, We aim at providing several key findings when distilling homogenous representations in this work. More sophisticated techniques for feature and instance distillation are possible, however, we intend to use simple approaches, so that our method can be easily applied to various multi-view 3D detectors, with minimum modifications to the original models.
>
> Q2: The difference between it and UVTR, and why it offers more opportunities for the network to align information.
>
> Sorry for your confusion. For most 3D detectors, such as CenterPoint and SECOND, the BEV features denote the features after the 3D backbone (and the FPN neck, if possible), where the voxelized features are flattened along z-axis to form the BEV feature maps. UVTR and LIGA-Stereo, directly utilize this feature to distill the image-based detector. However, in our implementation, we choose the features after the DETR encoder, as illustrated in Figure 1 top branch. With the effective self-attention operation, the BEV features provided by the teacher model contain more global context information, which can be more beneficial to the cross-modal feature imitation according to our experiments. We have added more explanation in dense feature distillation part to make it more clear in our updated paper.
>
> Q3: Why the BEV feature in Figure 1 is 3D BEV?
>
> Sorry for your confusion. We simply want to differentiate the BEV features in the 2D branch and 3D branch, 2D BEV indicates the features generated from 2D images, and 3D BEV indicates the features generated from 3D point clouds. They are both BEV features, without height information. We have updated `3D BEV` and `2D BEV` to `BEV Feature` to avoid ambiguity in our updated paper.
>
> Q4: It's better to report Table 10 in the main paper.
>
> Thanks for your suggestion. We have updated our paper to move Table 10 into the main paper in our new version.

---

### Official Review · Reviewer_BsUP · 2022-10-24

**Confidence:** 4
**Correctness:** 3
**Technical Novelty And Significance:** 2
**Empirical Novelty And Significance:** Not applicable
**Recommendation:** 6

**Clarity, Quality, Novelty And Reproducibility:**

The method is clearly presented and easy to understand, but some necessary implementation details are missing, which affects its reproducibility. The distillation setting is novel but the technical novelty of the proposed method is limited (see weaknesses above).

**Strength And Weaknesses:**

Pros)
Distilling knowledge from LiDAR-based detectors to multi-view image-based detectors is a novel attempt and can be a practical technique to enhance image-based detectors.
The effectiveness of the method and its components is shown by the experiments.

Cons)
The designed IoU-based quality-score itself is not technically novel (explored in LiDAR-based methods [1,2]), and its effectiveness is not analyzed in the experiments.
Although effective, mutual information-based knowledge distillation is also studied in previous works [3]. This paper directly used an existing formulation.
Some necessary implementation details are missing, e.g., the overall loss and the hyper-parameters (\alpha and \beta in Eq.(13), \sigma_i in Eq(2), and \gamma in Eq(6)) are not given. These could affect the reproducibility of the proposed method.

[1] Zheng W, Tang W, Chen S, Jiang L, Fu CW. Cia-ssd: Confident iou-aware single-stage object detector from point cloud. AAAI 2021.
[2] Hu Y, Ding Z, Ge R, Shao W, Huang L, Li K, Liu Q. Afdetv2: Rethinking the necessity of the second stage for object detection from point clouds. AAAI 2022.
[3] Tian Y, Krishnan D, Isola P. Contrastive representation distillation. ICLR 2020.


**Summary Of The Paper:**

This paper proposes a method named BEVDistill to distill knowledge from LiDAR-based 3D object detectors (teacher) to enhance multi-view 3D detectors (student).
First, since both detectors project their inputs to the BEV space, the proposed method performs feature distillation between the BEV feature maps under the guidance of a ground-truth gaussian mask.
Second, the teacher and student both adopt DETR-style prediction heads, the prediction-level distillation requires a matching between the two sets of instance predictions generated by the teacher and student models. Instance-level distillation is then performed between each pair of matched instances through a bounding box regression loss and a mutual information loss between the teacher and student’s hidden representations before logits. To suppress low-quality instance predictions, the authors also design an IoU-weighted quality score as the weight for each instance pair in the distillation loss.

The experiments on the nuScenes dataset proves that the proposed method works well for the chosen student model (BEVFormer) with various backbones. The ablation studies demonstrated the effectiveness of feature and instance distillation, as well as several specific designs within each component.


**Summary Of The Review:**

Although this paper proposed an effective method for distilling knowledge from LiDAR-based detectors to multi-view image-based detectors, which is a novel attempt towards enhancing image-based detectors, the technical novelty of each proposed component is limited, since IoU reweighting and mutual information-based KD are previously studied. The experiments demonstrate reasonable improvements but some necessary implementation details are missing. Overall, this paper is marginally below the acceptance threshold.

---

> ### Author Response · Authors · 2022-11-13
> **Response to Reviewer BsUP**
>
> Thanks for your valuable comments on our paper.
>
> Q1: The designed IoU-based quality-score and mutual information-based knowledge distillation are not technically novel.
>
> The main intuition of our paper is to explore an effective framework of cross-modal distillation for multi-view 3D object detection. Our method aims at dealing with different representations across student and teacher models, with simple yet effective solutions. More sophisticated techniques for feature and instance distillation are possible, however, we intend to use simple approaches, so that our method can be easily applied to various multi-view 3D detectors, with minimum modifications to the original models. Besides, these simple techniques are based on our key findings: (i) point-based BEV features hold different representations compared with image-based BEV features (ii) logit-based distillation is not suitable for cross-modal knowledge distillation due to the differences across non-homogenous inputs, which are non-trivial.
>
>
> Q2: The reproducibility of the proposed method.
>
> Sorry for your confusion on our implementation. The overall loss is a combination of the original 3D detection loss, the dense feature distillation loss (Sec.3.2.1), and sparse instance distillation loss (Sec.3.2.2). The $\sigma$ in Eq(2) is set to 2, $\gamma$ in Eq(6) is set to 0.1, $\alpha$ and $\beta$ in Eq(13) are the same with the original loss weight, 1.0 and 0.25, respectively. We have updated the hyper-parameters clarification in our updated paper and we will release the code for future research and ease others to reproduce our work.

---

### Official Review · Reviewer_zDXj · 2022-10-25

**Confidence:** 3
**Correctness:** 4
**Technical Novelty And Significance:** 2
**Empirical Novelty And Significance:** 3
**Recommendation:** 8

**Clarity, Quality, Novelty And Reproducibility:**

The method is clearly written, and would be reproducible by someone with some experience. At the time of the review, there does not seem to be any code released or promises of code release. The work is well motivated, of high quality, and easy to read. The greatest contribution of this work appears to be in the experimental results, and the strong performance on the nuScenes dataset shows the distillation proposed has potential. It appears to be built off of a few existing architectures, in particular the BEVFusion work, and novelty seems limited. The method of distillation is inspired by other works, but cleverly implemented by the authors. However, the strong experimental section is noteworthy and should be considered a strong contribution.

**Strength And Weaknesses:**

**Strengths**
The paper is very well written, and analytically thorough. The strongest point of this paper is the clear and complete experimental section, that exhaustively ablates the key components of the model. Reading over the tables, I observe that it is competitive as compared to a strong set of baselines (Table 1), is robust across backbones (Table 2), and is a well motivated distillation method with each component contributing to its performance (Table 3). Baseline wise, the authors compare against a strong set of methods, that – to the best of my knowledge – is comprehensive. Further ablations on the method’s distillation components and losses are interesting and informative, and convincingly beat a set of baseline comparisons.

One interesting point is that this method cleverly deals with different representations between the Student and Teacher models, which doesn’t limit the architecture of both (up until the distillation representation). This distillation process (foreground focusing) is intuitive and is validated experimentally (Table 4).


**Weakness**
One weakness is that the method appears to only be evaluated on a single dataset, nuScenes. It is strange to me, however, since the model BEVfusion from which they are building off of is evaluated both on nuScenes dataset and Waymo dataset. **This paper would be much stronger if there is a second dataset that they compare their method on**. This would ensure that their results – while excellent – are also general.

**Misc**
I am also curious if this distillation process (across BEV features) works on different BEV type detectors? It is a minor point, but does this distillation improve other Pseudo-LiDAR (Wang 2019) type methods, Lift-Splat (Philion, 2020) type methods, or Inverse Perspective Mapping (Reiher, 2020)? In theory, the method should improve upon all of the detectors that utilize intermediate BEV representations, and I would be curious to see if this is indeed the case.


**Summary Of The Paper:**

This paper explores LiDAR-to-Monocular distillation methods for BEV object detection. Their method, BEVDistill, builds upon the work of BEVFusion and uses the intermediate BEV representation to ease the multi-modal alignment and feature imitation. This allows the use of an asymmetrical LiDAR detector, Object-DGCNN, which this paper chose to use. The method also uses a “dense feature distillation” and “sparse instance distillation”, which supervises both the feature level (with a soft-focus on the foreground region) and the instance level (focusing on high quality teacher predictions). This method is evaluated extensively on the nuScenes dataset, with a strong leaderboard performance (5th place on NDS score at the time of review). The authors also conduct extensive ablation studies to validate the components of the model.

**Summary Of The Review:**

In summary, this work builds upon the BEVFusion work and utilizes the intermediate BEV representation to ease the distillation process between detectors using LiDAR input to detectors using monocular camera input. By using their  “dense feature distillation” and “sparse instance distillation”, they are able to achieve very strong performance on the nuScenes dataset. The strong analysis section and ablations support their claim. However, their results are slightly limited by the fact that their results are only on a single dataset. Overall, the work is clear; I would be willing to increase the score if additional results on another dataset support their claims.

---

> ### Author Response · Authors · 2022-11-13
> **Response to Reviewer zDXj**
>
> We sincerely appreciate your constructive and very detailed comments.
>
> Q1: Experiments on another dataset to ensure the generalization of the proposed method.
>
> Sorry for your confusion. Actually, most current multi-view 3D object detectors only report their performances on nuScenes dataset in their official paper, (e.g. BEVFusion, PETR-series, BEVDet-series, BEVDepth, UVTR). Though BEVFusion achieves great performance on the Waymo dataset, they do not release their implementations on it. Therefore, it's hard to compare them fairly. Due to the time limitation, we experiment BEVDistill on another strong multi-view 3D detection baseline: MV-FCOS3D++, (the second place in the Waymo Open Dataset competition, Camera-Only Track) in Table S2, which proivdes their codes on the Waymo Open Dataset. We adopt SECOND as our teacher model to adapt with the MV-FCOS3D++ as student model. It can be seen that BEVDistill can consistently improve MV-FCOS3D++ and MV-FCOS4D++ (multi-frame input version of MV-FCOS3D++) by 1.7/1.1 mAPL, respectively, further validating the effectiveness of BEVDistill. Detailed experimental settings can refer to the updated paper in Appendix B.4.
>
> |Model|Backbone|BEVDistill|mAPL|mAP|mAPH|
> |-|-|:-:|-|-|-|
> |MV-FCOS3D++|ResNet-50||32.6|45.1|42.7|
> |MV-FCOS3D++|ResNet-50|$\checkmark$|34.3|46.8|44.4|
> |MV-FCOS4D++|ResNet-50||34.0|46.5|43.8|
> |MV-FCOS4D++|ResNet-50|$\checkmark$|35.1|47.5|44.9|
>
> *Table S2: Experimental results on MV-FCOS3D++ and MV-FCOS4D++ on ResNet-50 w and w/o BEVDistill. Due to time limitations, all models are trained on the 1/5 Waymo training subset with 24 epochs and evaluated on the validation subset.*
>
> Q2: If BEV distillation can work on different BEV-type detectors?
>
> It is an interesting topic to explore the distillation process on various `cam2bev` types. Since current detectors are mainly based on LSS, pseudo-lidar, and cross-attention, we select BEVDet, BEVDepth, and BEVFormer as representative detectors to investigate the effectiveness of BEV distillation. The results are shown in Table S3. It can be concluded that BEV distillation can enhance performance on all kinds of detectors, by 1.9, 1.2, and 2.4 NDS, which proves that such a distillation paradigm is quite robust to the type of various `cam2bev` operations.
>
> |Model| `cam2bev` Type | BEVDistill| mAP|NDS|
> |-|:-:|:-:|-|-|
> |BEVDet|LSS||28.7|39.1|
> |BEVDet|LSS|$\checkmark$|30.5|41.0|
> |BEVDepth|Pseudo-Lidar||31.7|44.0|
> |BEVDepth|Pseudo-Lidar|$\checkmark$|33.0|45.2|
> |BEVFormer|Cross-Att||29.2|41.1|
> |BEVFormer|Cross-Att|$\checkmark$|31.0|43.5|
>
> *Table S3: Experimental results on various cam2bev 3D detectors w and w/o BEVDistill on nuScenes dataset. Due to time limitations, all models are trained on the 1/2 nuScenes training subset with 24 epochs and evaluated on the validation subset.*

---

### Official Review · Reviewer_2Rwe · 2022-10-25

**Confidence:** 4
**Correctness:** 3
**Technical Novelty And Significance:** 3
**Empirical Novelty And Significance:** 3
**Recommendation:** 6

**Clarity, Quality, Novelty And Reproducibility:**

Good clarity and quality.
The novelty is accecptable.
The reproducibility is good since sufficient details (e.g., proofs, experimental setup) are described.

**Strength And Weaknesses:**

Strength
+ The overall design is relatively novel,  it unifies the image and LiDAR features in the BEV space and adaptively transfer knowledge across non-homogenous representations in a teacher-student paradigm.
+ The method is well ablated and  achieves state-of-the-art performance on nuScenes.
+ The writing is good and easy to follow.

Weakness
- It would be better if some visualizations are provided to verify the effectiveness of proposed method.
- More experiments on comparison with BEVDepth will be interesting, which directly uses point clouds as depth ground truth and supervises the image detector. This comparison is necessary to clarify the motivation of the teacher-student paradigm, since it is also possible to supervise the image detector with LiDAR point cloud (more specifically, depth).

**Summary Of The Paper:**

This paper proposes a cross-modal knowledge distillation framework for mutli-view 3D object detection, with a 3D point cloud detector as teacher and an image detector as student.  The proposed method unifies different modalities in the BEV space and conducts knowledge distillation through the dense feature distillation and the sparse instance distillation. Extensive experiments demonstrate that the proposed method outperforms current KD approaches, and the method improves baseline model with a significant boost and achieves new state-of-the-art performance on nuScenes test leaderboard.

**Summary Of The Review:**

The paper presents a cross-modal knowledge distillation framework for mutli-view 3D object detection which unifies the image and LiDAR features in the BEV space. The paper has good writing and extensive experiments. However, the paper is lack of more visualizations to verify the proposed method.

My biggest concern is regarding the motivation of using the teacher-student distillation paradigm, since it is also possible to supervise the image detector with LiDAR point cloud (more specifically, depth). If authors can provide comparisons with BEVDepth and show the necessarity of the adopted distillation setting, I may raise my rating.

---

> ### Author Response · Authors · 2022-11-13
> **Response to Reviewer 2Rwe**
>
> Thanks for your valuable comments on our paper.
>
> Q1: Visualizations to verify the effectiveness of the proposed method.
>
> Thanks for your advice. We provided some visualizations in Appendix D. To further verify our approach, we add more visualizations on nuScenes and Waymo as videos in the updated supplementary materials. The ground truth and predictions are in orange and blue, respectively.
>
> Q2: More experiments on comparison with BEVDepth.
>
> Thanks for your kind suggestion. In order to explore the effectiveness of teacher-student paradigm compared to LiDAR (depth) supervision, we perform BEVDistill on a BEVDepth baseline without depth supervision (by setting the loss_depth_weight to 0) and the results are shown in Table S1. From the table, we can observe that the improvement brought by BEVDistill is larger than depth supervision (1.4 NDS vs 0.9 NDS), demonstrating the superiority of the teacher-student paradigm. Actually, depth supervision only brings 0.9 NDS enhancement, according to our experiments (43.1 NDS to 44.0 NDS) and the official BEVDepth paper (33.3 NDS to 34.2 NDS). What's more, our experiments also indicate that the teacher-student paradigm may not be in conflict with LiDAR supervision, since BEVDistill can still improve the performance of BEVDepth, which already utilizes LiDAR signals as direct supervision. Besides, depth supervision is inapplicable for those `cam2bev` approaches with implicit depth estimation view transformation, such as cross-attention (BEVFormer), while BEVDistill is agnostic to such operations. Overall, the teacher-student paradigm provides a general and extendable solution for leveraging LiDAR signals for multi-view 3D object detection.
>
> |Model| depth supervision|BEVDistill|mAP|NDS|
> |-|:-:|:-:|-|-|
> |BEVDepth|||30.5|43.1|
> |BEVDepth||$\checkmark$|32.1|44.5|
> |BEVDepth|$\checkmark$||31.7|44.0|
> |BEVDepth|$\checkmark$|$\checkmark$|33.0|45.2|
>
> *Table S1: Experimental results on BEVDepth w and w/o depth supervision and BEVDistill. Due to time limitations, all models are trained on the 1/2 nuScenes training subset with 24 epochs and evaluated on the validation subset.*

---

### Decision · Program_Chairs · 2023-01-20

**Decision:**

Accept: poster

**Justification For Why Not Higher Score:**

The proposed method is very specific and while effective, it is unclear whether it can be applied to other classes of problems.  Reviewers also noted that the technical novelty of the individual pieces are limited and have been explored in other work.

**Justification For Why Not Lower Score:**

Reviewers all found the paper to be well-written and the proposed approach sufficient novel, and to work well, and all recommend accept.

**Metareview: Summary, Strengths And Weaknesses:**

The paper proposes BEVDistill, a cross-modal knowledge distillation framework for multi-view 3D object detection where all features are transformed into BEV space.  BEVDistill uses a LiDAR-based 3D point cloud detector as the teacher and an multi-view image detector as the student model.  The method support both a "dense feature distillation", with weights on foreground regions from the teacher (as LiDAR points will be only for part of the image), and "sparse instance distillation", where high quality instances from the teacher are weighed more.  Experiments on nuScenes show that the proposed approach outperforms prior distillation approaches and there are extensive ablation studies to justify the various model components.

Strengths
- The proposed method of knowledge distllation from LiDAR to image in BEV space is sufficiently novel and works well
- Experiments show the proposed approach outperforms prior approaches on nuScenes (and also Waymo after rebuttal)
- Paper is well written with clear description of the method, and careful experiments and ablations

Weaknesses
- Reviewers noted that the technical novelty of each component is limited
- Reviewers also noted that there may be some missing details that could make some parts of the method not completely reproducible



**Note From Pc:**

if the above contains the word "oral" or "spotlight" please see: "oral" presentation means -> notable-top-5% and "spotlight" means -> notable-top-25%. As stated in our emails, we are disassociating presentation type from AC recommendations